# Robust Optimization of Cold Chain Logistics Networks with Time Window under Uncertain Demand: A Case Study in China

Jie Liao
*Business school*
*Sichuan University*
Chengdu, China
Email: liaojie@stu.scu.edu.cn

Yalan Li
*Business School*
*Sichuan University*
Chengdu, China
Email: liyalan719@163.com

Jiyang Liu
*Administrative Department*
*Sichuan New Energy Power Company Limited*
Chengdu, China
Email: 13438836393@163.com

*Abstract*—**China's cold chain industry suffers from the problems of imperfect cold chain logistics networks and irrational planning, causing high cold chain loss and high enterprise costs, and severely restricting the development of the industry. Considering the high uncertainty of cold chain market demand, this paper adopts the robust optimization method to construct an optimization model of a cold chain logistics network with a time window by taking the minimization of the comprehensive cost of the system as the goal, and uses a genetic algorithm to solve the model. This paper takes Company A as an example for analysis, and obtains the optimization scheme through the optimization model. The location of the optimal distribution center is obtained, and six distribution paths are planned so as to form a complete cold chain network system. Meanwhile, the optimization results under different demand fluctuation risk levels and different freight conditions are considered, and the results show that the location selection and path scheduling will not change under different demand fluctuation levels. In addition, a sensitivity analysis of the dynamic change of freight rate was carried out. When the freight rate reaches \$4.5/t/km, not only the transportation cost and cargo damage cost will increase accordingly, but also the routing scheme will change accordingly.**

*Keywords*—**Cold chain logistics; Time window; Location routing optimization; Demand uncertainty; Robust optimization**

## I. INTRODUCTION

The cold chain, a specialized logistics network, ensures the continuous maintenance of perishable goods in a low-temperature environment throughout their lifecycle, from pre-cooling to final sale. It is essential for preserving the health, freshness, and timeliness of delivered food products[1]. Accurate regulation of both time and temperature is imperative for guaranteeing the timely provision of high-

The work was supported by the National Natural Science Foundation of China [Grant No. 71771157], the Fundamental Research Funds for the Central Universities, Sichuan University [Grant No. SCJJ-14, SKSYL2021-05], Funding of Sichuan University [Grant No. skqx201726]. (Corresponding Author: Jiyang Liu)

quality, healthful perishables to customers[2]. Precise control of time and temperature is a prerequisite for ensuring the timely delivery of healthy, fresh, high-quality perishable products to In a vehicle-equipped logistics system, the temperature can be kept constant while driving. On the other hand, non-equipped transportation systems require at least two complementary operations, preservation, and freezing, to be applied in consolidated warehouses or hubs[3].

Currently, China's cold chain logistics industry still lags significantly behind developed nations, with high distribution costs constituting a significant barrier to its healthy development. Moreover, an insufficient and irrationally structured cold chain network leads to low logistical efficiency, further impeding the progress of this vital sector in China.

The integration of internal and external factors is pivotal in optimizing cold chain logistics systems. An integrated robust optimization model is proposed, addressing distribution center location and vehicle routing, to minimize costs. This model offers tailored solutions based on decision-makers' risk tolerance, facilitating quantitative analysis and data-driven decision-making. It enhances resilience to market fluctuations, boosts operational efficiency, optimizes system-wide costs, and fosters the development of economical, efficient modern logistics systems. Ultimately, it contributes to the long-term sustainability of the cold chain logistics industry, underscoring its theoretical and practical significance.

The main contributions of this study are as follows:

(1) An optimization model for cold chain logistics networks with time windows is formulated, addressing various influencing factors. It performs location-path planning, leverages transportation volume as a decision variable to analyze resource allocation and balance supply-demand, ultimately supporting managers' transportation decisions with data insights.

(2) The robust optimization model is used to solve the distribution center siting problem and vehicle path planning problem. It is a relatively innovative attempt to optimize the cold chain logistics network system under the influence of internal and external vertical and horizontal factors. which is superior to the traditional uncertainty optimization method. When the uncertain parameters fluctuate arbitrarily in the set, the model can come up with a satisfactory solution, which reflects the robustness of the model.

(3) Take company A as an example to evaluate the effectiveness of the established model and method.

The remainder of this article is organized as follows. In Section 2, the literature related to this study is reviewed. Section 3 describes our assumptions and the robust optimization model. Section 4 presents a numerical example to evaluate the validity and applicability of the model. We also perform sensitivity analysis and discuss the results. Finally, Section 5 presents conclusions and recommendations.

## II. RELATED WORK

### A. Cold Chain Logistics

Supply chain management involves managing production and delivery processes by considering the entire supply chain network from the beginning to the end of the product lifecycle[4][5][6]. As an important supply chain system, cold chain logistics solves the problem of low-temperature transportation and helps to improve food preservation(for example, fresh grapes[7][8] and salmon[9]. The quality and safety of food in cold chain are closely related to the three factors: the refrigeration equipment, the humidity and the temperature[10]. According to the status and performance of Canada's food cold chain, the use of GIS's risk assessment, simulation, and planning tools (CanGRASP) has opened up a way for the prevention of foodborne disease outbreaks and the study of new systems to reduce food waste and energy consumption[11]. With the development of many aspects of research, the research of cold chain technology is also constantly breaking through. A model of multi-temperature common distribution mode is used for food cold chain to realize the upgrading of logistics services, which greatly improves the competitiveness of the logistics industry in the field of thermal protection of fragile products and temperature-sensitive goods[3]. Some researchers have summarized the world's research results in promoting the intelligence of food logistics, focusing on the changes in the shelf life of products monitored by technology, models, and applications[12]. In terms of storage technology, better quality monitoring, shelf-life biological models, and FEFO adjustments to reduce losses and improve quality[13][14].

### B. Distribution Center Location and Vehicle Routing Planning

Distribution center location problem and vehicle path planning problem are two key hubs in logistics network optimization. These two hubs interact and coordinate with each other. scholars took the location-routing as a whole for collaborative research. Focusing on the research of node location optimization decisions in the network, the multi-node location model, the optimal location-scale bi-level optimization model are constructed. When studying the location decision problem, considering the vehicle transportation path problem, a combinatorial optimization model of the location-routing problem is constructed [15][16][17][18]. Some scholars introduced the time window factor into the model, and used the branch and bound exact algorithm to solve the model, then conducted sensitivity analysis for the time window[19]. Considering the inventory factor in the location-routing problem, a three-stage supply chain model of location-inventory-routing is established, and a two-stage heuristic algorithm is selected to solve the problem. In the second stage, the simulated annealing algorithm is applied to improve the results[20]. Yan et al. established a two-stage distribution location-routing model with cost minimization as the objective function and vehicle capacity changes at different distribution stages[21].

### C. Uncertain Factors in Cold Chain Logistics

As academic research increasingly integrates with practice, scholars now incorporate real-world uncertainties into their studies. Research on uncertain problem optimization focuses on stochastic programming, fuzzy programming, and robust optimization. Most scholars aim to minimize total cost, shorten paths, or delivery times, integrating distribution center location with path planning to form a comprehensive three-tier supply chain network. However, despite the prevalence of uncertainties due to natural factors, market demand fluctuations, and industry competition, research addressing these uncertainties remains limited. In cold chain logistics, uncertainty research mainly focuses on emergency logistics, often employing stochastic and fuzzy programming. However, stochastic programming faces challenges in predicting random variable distributions, while fuzzy programming's subjectivity limits its applications. Given China's rapid cold chain market expansion, the market now drives industry development. Under the condition of uncertain demand, this paper introduces robust optimization to solve the integration problem of distribution center location and transportation path planning in a cold chain logistics system. At the same time, in order to ensure the maximum satisfaction of the timeliness requirements of cold chain products, time window penalty cost is added. That is, by establishing a robust optimization model of a cold chain logistics network with time window, the location and number of distribution centers are determined, and the transportation path is planned to meet the needs of each demand point, so as to achieve the goal of the lowest cost of the whole system.

## III. METHODS

Based on the problem description and reasonable research hypothesis, the comprehensive cost of the whole system of the cold chain logistics network is analyzed, the uncertainty of market demand is fully considered, and the minimum system cost is taken as the objective function. A robust optimization model of the cold chain logistics network with time window is constructed. The structure of the three-level cold chain network system is shown in "Fig. 1". This kind of complex combination problem belongs to an NP-hard problem, and there are some difficulties in solving it. In this paper, the genetic algorithm is used to design the solution program for the model through MATLAB R2018 b.

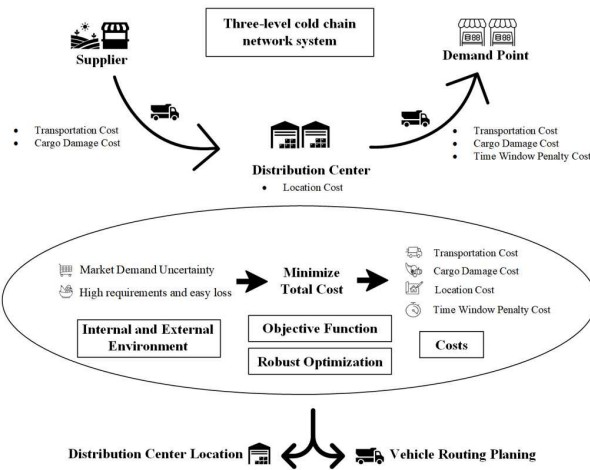

Fig. 1.   Structure of the three-level cold chain network system

## A. Hypothesis

(1) The location of the supplier and the demand point is determined and known, and the supplier's volume is sufficient to meet the demands.

(2) Vehicles sent from the distribution center will return to the starting point after the task is completed, and only one vehicle will be arranged for distribution on each line.

(3) The requirements of each demand point are independent, and each demand point can only be satisfied by one distribution center.

(4) In the whole cold chain system, the vehicle from the supplier to the distribution center has the same load and model, and is served by one type of vehicle, which is transported at a constant speed and the speed is known.

(5) The load and model of the vehicle from the distribution center to the demand point are also the same. It is served by one type of vehicle, transported at a constant speed and the speed is known.

(6) The transportation process is constant temperature transportation and the damage rate is fixed and known. The cost of cargo damage is only related to the transportation time.

(7) Delivery is strictly according to the soft time window limit, early delivery or late delivery will pay the penalty cost, and the penalty cost is linearly related to the time of early delivery or late delivery.

## B. Decision Variable and Parameters

$X_i$ : The amount of fresh products transported from the supplier to the distribution center $DC_i$ .

$Y_{ij}$ : The amount of fresh products delivered from $DC_i$ to retail point $j$.

$$DC_i = \begin{cases} 1 & \text{If the distribution center i is selected} \\ 0 & \text{Else} \end{cases}$$

$$R_{ij} = \begin{cases} 1 & \text{If the demand of the retail point j is met by the distribution center i} \\ 0 & \text{Else} \qquad\qquad i \in I, j \in J \end{cases}$$

$$r_{gjk} = \begin{cases} 1 & \text{If the vehicle opens from node g to node j} \\ 0 & \text{Else} \qquad\qquad \forall g \in (I \cup J), j \in J \end{cases}.$$

The meaning of the relevant parameters in the model construction is shown in Table Ⅰ.

TABLE Ⅰ
PARAMETERS.

| Parameter | Description |
|---|---|
| $I$ | Aggregation of distribution centers |
| $J$ | Aggregation of customer demand points |
| $K$ | Aggregation of vehicles |
| $D_j$ | The demand at customer demand point j |
| $C_i$ | Distribution center location cost for center i |
| $C_t$ | Transportation costs per unit of product per unit distance |
| $C_s$ | Cost of goods lost per unit of product |
| $d_i$ | Distance from supplier to distribution center $DC_i$ |
| $d_{gi}$ | Transportation distance from node g to node j |
| $V_i$ | Speed of transportation from suppliers to distribution centers |
| $V_{gj}$ | Distribution speed from node g to node j |
| $\theta$ | Rate coefficient for spoilage of products in transit |
| $\Gamma$ | Robustness factor, $\Gamma \in [0,1]$ |
| $E_j$ | Upper limit of the time window |
| $L_j$ | Lower limit of the time window |
| $T_{jk}$ | The time point when vehicle k arrives at node j for delivery. |
| $T_{gk}$ | The time point when vehicle k arrives at node g for delivery. |
| $ST_{gk}$ | Time period during which vehicle k stays at node g |
| $P_{de}$ | The penalty coefficient by the early arrival of the delivery |
| $P_{dl}$ | The penalty coefficient for late delivery |
| $Q_i$ | Maximum capacity limit for distribution center i |
| $L_k$ | Maximum limits of carriage of vehicle k |
| $\bar{D}_j$ | The nominal value of the retail point demand j |
| $\hat{D}_j$ | The constant perturbation of the retail point j |
| $\xi$ | The disturbance coefficient, $\xi \in [-1,1]$ |

## C. Modeling

This paper aims to minimize the total cost of a three-level cold chain logistics network system. In contrast to general logistics, cold chain logistics for fresh products encounters

challenges such as product damage due to temperature fluctuations and extended transportation times. Addressing these issues necessitates refined transportation route planning and temperature control, thereby increasing costs. Consequently, the comprehensive cost of the cold chain logistics system encompasses not only traditional location and transportation expenses but also accounts for cargo damage costs and penalties for breaching distribution time windows. The costs of each component are detailed as follows:

(1) Distribution center location cost: Location costs encompass warehouse construction, land lease, equipment purchase, maintenance, and operation expenses. The total distribution center location cost is expressed as:

$$f_1 = \sum_i (c_i \times DC_i) \tag{1}$$

(2) Vehicle transportation cost: We divide the cold chain into two stages of transportation. Assuming that the unit transportation costs of the two stages are the same, the first stage is that the supplier transports fresh products to the distribution center for transshipment, packaging, circulation processing and other pre-distribution activities. The transportation cost can be expressed as $\sum_i C_t X_i d_i DC_i$. The second stage is to distribute the demand of the demand point from the distribution center, and the transportation cost can be expressed as $\sum_{g \in (I \cup J)} \sum_j C_t Y_{gj} d_{gj} r_{gjk}$. Then the total transportation cost can be expressed as:

$$f_2 = \sum_i C_t X_i d_i DC_i + \sum_{g \in (I \cup J)} \sum_j C_t Y_{gj} d_{gj} r_{gjk} \tag{2}$$

(3) Cost of cargo damage: The main source of cargo damage in cold chain transportation is the damage to the product quality caused by long transportation time and the damage caused by unsaleable products caused by excessive supply. The number of fresh agricultural products at the initial shipment is $z_{ij}(0)$. Taking this as a standard to calculate the amount of loss generated during the transportation process, it is assumed that the cold chain fresh products are corrupted at a constant rate during transportation, the cargo damage differential equation is: $dz_{ij}(t)/d_t = -\theta z_{ij}(0)$, The cargo damage equation is: $z_{ij}(T) = x_{ij} e^{-\theta sij/vij}$. The cargo damage cost of the three-level supply chain cold chain logistics system is divided into two stages: the first stage is the damage cost from the supplier to the distribution center is $\sum_i C_s X_i (1 - e^{-\theta d_i/v_i})$, the second stage is the cost of damage from the distribution center to the retail point is $\sum_j \sum_g C_s Y_{gj} (1 - e^{-\theta d_{gj}/v_{gj}})$. The damage cost is:

$$f_3 = \sum_i C_s X_i (1 - e^{-\theta d_i/v_i}) + \sum_{g \in (I \cup J)} \sum_j C_s Y_{gj} (1 - e^{-\theta d_{gj}/v_{gj}}) \tag{3}$$

(4) Penalty cost for violation of time window: In the cold chain transportation of fresh food, soft time window is used to describe the penalty cost beyond the time window. The upper

and lower limits of the time window of the retail point j are represented as $[E_j, L_j]$, the transportation time between two nodes is equal to the ratio of the distance between two points to the speed, which can be expressed as $d_{gj}/V_{gj}$. Then, the penalty cost beyond the time window can be expressed as:

$$f_4 = P_{de} \sum_j \max(E_j - T_{jk}) + P_{dl} \sum_j \max(T_{jk} - L_j, 0)$$
$$T_{jk} = (T_{gk} + ST_{gk} + d_{gj}/V_{gj}) \times r_{gjk} \quad g \in (I \cup J), j \in J, k \in K. \tag{4}$$

The constraints are as follows:

At least one distribution center was selected:

$$\sum_i DC_i \geq 1 \tag{5}$$

Distribution center capacity constraints, the number of purchases from suppliers is less than the maximum capacity of the distribution center:

$$X_i \times DC_i \leq Q_i \quad i \in I \tag{6}$$

Logistics vehicle carrying capacity constraints, the number of products transported within the maximum capacity limit of the vehicle:

$$\sum_j Y_{ij} \sum_j r_{gjk} \leq L_k \quad i \in I, g \in (I \cup J) \tag{7}$$

Each retail outlet has only one car to serve it:

$$\sum_{g \in (I \cup J)} \sum_k r_{gjk} = 1 \quad j \in J \tag{8}$$

Each vehicle serves at most one distribution center:

$$\sum_{g \in (I \cup J)} \sum_j r_{gjk} \leq 1 \quad k \in K \tag{9}$$

The vehicle cannot stay at a certain node and needs to return to the departure distribution center:

$$\sum_{g \in (I \cup J)} r_{gjk} - \sum_{g' \in (I \cup J)} r_{jg'k} = 0 \quad j \in J, k \in K \tag{10}$$

In the first stage of transportation, the fresh product volume from the supplier to the distribution center should exceed the sum of product in the second stage and losses in the first stage:

$$X_i \geq \sum_j Y_{ij} + X_i (1 - e^{-\theta d_i/v_i}) \quad i \in I \tag{11}$$

In the second stage of transportation, the total distribution from the center must satisfy retail point demands and account for transportation losses:

$$\sum_{j} Y_{ij} \geq \sum_{g \in (I \cup J)} \sum_{j} D_j + Y_{gj}(1 - e^{-\theta d_{gj}/v_{gj}}) \quad i \in I \qquad (12)$$

The time for vehicle $k$ to reach the retail point $j$ of fresh agricultural products is equal to the sum of the time for vehicle $k$ to reach the previous node $g$ plus the time spent at point $g$ and the time spent on the transportation of the vehicle from point $g$ to point $j$:

$$T_{jk} = (T_{gk} + ST_{gk} + d_{gj}/V_{gj}) \times r_{gjk} \quad j \in J, g \in (I \cup J), k \in K \qquad (13)$$

The distribution centers are independent of each other:

$$r_{abk} = 0 \quad a,b \in I, k \in K \qquad (14)$$

The non-negative constraints of decision variables:

$$
\begin{aligned}
DC_i &= \{0,1\} & i \in I \\
Rij &= \{0,1\} & i \in I, j \in J \\
r_{gjk} &= \{0,1\} & j \in J, g \in (I \cup J), k \in K \\
X_i &> 0, Y_{ij} > 0 & i \in I \ j \in J
\end{aligned}
\qquad (15)
$$

In summary, the three-level cold chain logistics transportation network optimization mathematical model is:

$$
\begin{aligned}
Obj \quad &\min F = f_1 + f_2 + f_3 + f_4 \\
&f_1 = \sum_i (c_i \times DC_i) \\
&f_2 = \sum_i C_t X_i d_i DC_i + \sum_{g \in (I \cup J)} \sum_j C_t Y_{gj} d_{gj} r_{gjk} \\
&f_3 = \sum_i C_s X_i (1 - e^{-\theta d_i/v_i}) + \sum_j \sum_g C_s Y_{gj}(1 - e^{-\theta d_{gj}/v_{gj}}) \\
&f_4 = P_{de} \sum_j \max(E_j - T_{jk}, 0) + P_{dl} \sum_j \max(T_{jk} - L_j, 0)
\end{aligned}
$$

s.t.

$$
\begin{cases}
\sum_i DC_i \geq 1 \\
X_i \times DC_i \leq Q_i \quad i \in I \\
\sum_j Y_{ij} \sum_j r_{gjk} \leq L_k \quad i \in I, g \in (I \cup J) \\
\sum_{g \in (I \cup J)} \sum_k r_{gjk} = 1 \quad j \in J \\
\sum_{g \in (I \cup J)} \sum_j r_{gjk} \leq 1 \quad k \in K \\
\sum_{g \in (I \cup J)} r_{gjk} - \sum_{g' \in (I \cup J)} r_{jg'k} = 0 \quad j \in J, k \in K \\
X_i \geq \sum_j Y_{ij} + X_i(1 - e^{-\theta d_i/v_i}) \quad i \in I \\
\sum_j Y_{ij} \geq \sum_{g \in (I \cup J)} \sum_j D_j + Y_{gj}(1 - e^{-\theta d_{gj}/v_{gj}}) \quad i \in I \\
T_{jk} = (T_{gk} + ST_{gk} + d_{gj}/V_{gj}) \times r_{gjk} \quad j \in J, g \in (I \cup J), k \in K \\
r_{abk} = 0 \quad a,b \in I, k \in K \\
DC_i = \{0,1\} \quad i \in I \\
Rij = \{0,1\} \quad i \in I, j \in J \\
r_{gjk} = \{0,1\} \quad j \in J, g \in (I \cup J), k \in K \\
Xi > 0, Yij > 0 \quad i \in I \ j \in J
\end{cases}
\qquad (16)
$$

## D. Dualistic Transformation

Retail outlets face high demand uncertainty due to external variability and consumer habits. To enhance planning accuracy, balance supply and demand, reduce waste and loss rates, improve capacity utilization, and boost economic benefits, it's crucial to integrate demand uncertainties into planning. This involves modeling demand as uncertain elements. The specific model transformation is as follows:

$$
U_u = \begin{cases}
D_j = \bar{D}_j + \xi \hat{D}_j \\
D_j \in [\bar{D}_j - \hat{D}_j, \bar{D}_j + \hat{D}_j] \\
|\xi| \leq \Gamma \\
\xi \in [-1,1]
\end{cases}
\qquad (17)
$$

The first is the representation of uncertain sets. In robust optimization, the results obtained by selecting different uncertain sets are also different. In this paper, the box uncertainty set is introduced, which has a linear structure and is easy to control the degree of uncertainty. It is widely used in practice. The uncertainty set of the model $U_u$ can be expressed in the following form.

Based on the theory of robust optimization, the uncertain factors in the model are represented by a linear combination of the nominal value of the demand and the constant disturbance, that is $D_j = \bar{D}_j + \xi \hat{D}_j$. The term with uncertainty coefficient is separated from the formula $\sum_j Y_{ij} \geq (\bar{D}_j + \xi \hat{D}_j) + Y_{gj}(1 - e^{-\theta d_{gj}/v_{gj}})$, In order to ensure the establishment of hard constraints, the demand will be taken $\max(\xi \hat{D}_j)$. The term with uncertain coefficients $\max(\xi \hat{D}_j)$ is equivalent to the following linear optimization model:

$$
\max(\xi \hat{D}_j) \\
\text{s.t.} \begin{cases} \xi \leq \Gamma \\ 0 \leq \xi \leq 1 \end{cases}
\qquad (18)
$$

By introducing the dual parameter $z_1$, $z_2$, the above formula is transformed into its dual form, and the uncertain coefficients in the model are transformed by the dual transformation, and the model is transformed into a deterministic type. The new dual form can be expressed as:

$$
\min(\Gamma z_1 + z_2) \\
\text{s.t.} \begin{cases} z_1 + z_2 \geq \hat{D}_j \\ z_1 \geq 0 \\ z_2 \geq 0 \end{cases}
\qquad (19)
$$

The complete three-level cold chain logistics network transportation robust optimization model is expressed as formula (20).

$$Obj \quad \min F = f_1 + f_2 + f_3 + f_4$$

$$f_1 = \sum_i (c_i \times DC_i)$$

$$f_2 = \sum_i C_t X_i d_i DC_i + \sum_{ig} Y_{ig} C_t d_{ig} R_{ij} r_{gjk}$$

$$f_3 = \sum_i C_s X_i (1 - e^{-\theta d_i / v_i}) + \sum_j \sum_g C_s Y_{gj} (1 - e^{-\theta d_{gj} / v_{gj}})$$

$$f_4 = P_{de} \sum_j \max(E_j - T_{jk}, 0) + P_{dl} \sum_j \max(T_{jk} - L_j, 0)$$

$s.t.$

$$
\begin{cases}
\sum_i DC_i \geq 1 \\[4pt]
X_i \times DC_i \leq Q_i \quad i \in I \\[4pt]
\sum_j Y_{ij} \sum_j r_{gjk} \leq L_k \quad i \in I, g \in (I \cup J) \\[4pt]
\sum_{g \in (I \cup J)} \sum_k r_{gjk} = 1 \quad j \in J \\[4pt]
\sum_{g \in (I \cup J)} \sum_j r_{gjk} \leq 1 \quad k \in K \\[4pt]
\sum_{g \in (I \cup J)} r_{gjk} - \sum_{g' \in (I \cup J)} r_{jg'k} = 0 \quad j \in J, k \in K \\[4pt]
X_i \geq \sum_j Y_{ij} + X_i (1 - e^{-\theta d_i / v_i}) \quad i \in I \\[4pt]
\sum_j Y_{ij} \geq \sum_{g \in (I \cup J)} \sum_j \overline{D}_j + (\Gamma z_1 + z_2) + Y_{gj} (1 - e^{-\theta d_{gj} / v_{gj}}) \quad i \in I \\[4pt]
z_1 + z_2 \geq \hat{D}_j \\[4pt]
z_1 \geq 0 \\[4pt]
z_2 \geq 0 \\[4pt]
T_{jk} = (T_{gk} + ST_{gk} + d_{gj} / V_{gj}) \times r_{gjk} \quad j \in J, g \in (I \cup J),, k \in K \\[4pt]
r_{abk} = 0 \quad a, b \in I,, k \in K \\[4pt]
DC_i = \{0,1\} \quad i \in I \\[4pt]
Rij = \{0,1\} \quad i \in I, j \in J \\[4pt]
r_{gjk} = \{0,1\} \quad j \in J, g \in (I \cup J), k \in K \\[4pt]
Xi > 0, Yij > 0 \quad i \in I \quad j \in J
\end{cases}
$$

$$(20)$$

## IV. CASE STUDY

In order to verify the practicability and effectiveness of the robust optimization model of cold chain logistics network with time windows constructed in this paper, due to the confidentiality of the actual enterprise strategic data, based on some actual enterprise background research, this paper introduces a virtual case company A, and uses its data to simulate and solve the model, so as to obtain a feasible cold chain logistics network optimization solution.

Company A is a domestic cold chain enterprise that deals mainly with fresh products. It mainly sells goods that need to be refrigerated and kept fresh. With the gradual development of enterprises, through the investigation and research of the expert group, it is found that M city is suitable for enterprises to carry out market expansion, and company A intends to carry out market scale expansion in Kaesong, M city. After preparation, the enterprise selected a supplier at (103.95, 30.8), three potential distribution centers, and identified 12 consumer collection points. While supplier, distribution center, and demand point locations are known, consumer demand is fuzzy and uncertain.

Therefore, according to the demand of company A, this paper uses the cold chain logistics network optimization model with time window to locate the distribution center and optimize the vehicle path planning. At the same time, the robust optimization method is introduced to solve the uncertainty of demand and provide quantitative analysis support for the decision scheme made by the enterprise.

### A. Datas

To expand its market and enhance competitiveness, Company A selected three distribution centers (DC1, DC2, DC3) in M city for potential use. At least one center must be chosen for short-term storage, transshipment, packaging, circulation, and processing of fresh products, with distribution arranged for 12 high-demand consumption points. The goal is to minimize total cold chain logistics cost while meeting each demand point's needs. The 12 consumer groups are numbered 1, 2, 3, etc. The related data of the 12 consumption points are shown in Table II. The coordinates, capacity and location costs of the three distribution centers are known, as shown in Table III. The other parameters of the model are shown in Table IV.

TABLE II
DEMAND POINT DATA.

| Number | axis(X) | axis(Y) | $\overline{D}_j$ | $E_j$ | $L_j$ | $ST_{gk}$ |
|---|---|---|---|---|---|---|
| 1 | 104.0336 | 30.6316 | 1.7849 | 20 | 45 | 20 |
| 2 | 104.0767 | 30.6237 | 1.1470 | 15 | 55 | 20 |
| 3 | 104.0914 | 30.6431 | 1.8067 | 10 | 40 | 20 |
| 4 | 104.0554 | 30.5499 | 0.9561 | 15 | 60 | 20 |
| 5 | 104.0332 | 30.4924 | 1.9068 | 45 | 95 | 20 |
| 6 | 103.8564 | 30.6881 | 1.8205 | 18 | 50 | 20 |
| 7 | 104.0952 | 30.6507 | 1.3050 | 50 | 100 | 20 |
| 8 | 104.0038 | 30.6706 | 1.6795 | 55 | 95 | 20 |
| 9 | 104.0592 | 30.6008 | 0.6534 | 15 | 55 | 20 |
| 10 | 104.1049 | 30.6701 | 1.8151 | 60 | 100 | 20 |
| 11 | 104.0465 | 30.5691 | 1.7808 | 50 | 100 | 20 |
| 12 | 104.0189 | 30.6888 | 1.1168 | 20 | 55 | 20 |

TABLE III
DISTRIBUTION CENTER DATA.

| Number | axis(X) | axis(Y) | Capacity(t) | Fixed cost(yuan) |
|---|---|---|---|---|
| DC1 | 103.9780 | 30.5462 | 15 | 3000 |
| DC2 | 104.0926 | 30.8125 | 15 | 2880 |
| DC3 | 104.0188 | 30.7511 | 15 | 2950 |

### B. Results and discussion

(1) Results under deterministic requirements

In this paper, MATLAB R2018 b is used to program and solve the model. The basic parameters in the genetic algorithm are set as follows: the population size is 200, the number of iterations is 2000, the probability Pc is 0.9, the mutation probability Pm is 0.05, and the generation gap is 0.9. The iterative optimization process is shown in Fig. 2.

As illustrated in Fig. 2, the objective function value converges to 13,227.2 yuan, representing an optimization from the initial value of approximately 15,500 yuan. Based on the path planning in Fig. 3, considering no demand fluctuation ($\Gamma$ =0), DC1 and DC3 are selected among the three alternatives. With distribution centers located, six paths and their respective traffic volumes are detailed in Table V. The total shipped products amount to 18.8529 tons, with a 0.5389-ton discrepancy due to in-transit losses, which is due to the loss of goods on the way from the supplier to the distribution center. Compared with the initial solution the obtained results achieve the purpose of optimization.

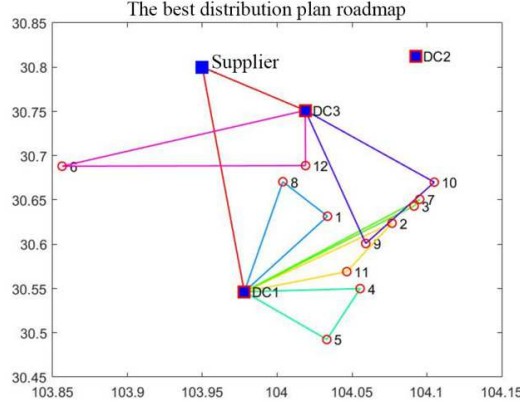

Fig. 2. Convergence diagram of genetic algorithm.

Fig. 3. Best location and distribution path planning diagram.

TABLE IV

MODEL PARAMETER VALUES.

| Parameter | Sign | Numerical value |
|---|---|---|
| Transportation cost per unit distance of agricultural products （yuan/t/km） | $C_2$ | 3 |
| Goods damage cost(yuan/t) | $C_3$ | 500 |
| Supplier-distribution center transportation speed(km/h) | $V_i$ | 80 |
| The transport speed from node g to node j(km/h) | $V_{gi}$ | 40 |
| Corruption rate coefficient of products in transit | $\theta$ | 0.05 |
| Robust coefficient | $\Gamma$ | [0,1] |
| Early arrival penalty coefficient | $P_{de}$ | 2 |
| late arrival penalty coefficient | $P_{dl}$ | 3.5 |
| The maximum carrying limit of vehicle k(t) | $L_k$ | 4 |
| Constant demand disturbance | $\hat{D}_j$ | 0.2826 |

TABLE V

OPTIMIZATION PATH RESULTS.

| Supplier | Transport volume (t) | Distribution center | Transportation path | Delivery volume (t) | Total cost (yuan) |
|---|---|---|---|---|---|
| S | 13.4379 | DC1 | 1: DC1->5->4-> DC1 | 3.0430 | 13227.2 |
| | | | 2: DC1->11->2-> DC1 | 3.1079 | |
| | | | 3: DC1->3->7->DC1 | 3.2917 | |
| | | | 4: DC1->1->8-> DC1 | 3.6444 | |
| | 5.9538 | DC3 | 5: DC3->10->9->DC3 | 2.6485 | |
| | | | 6: DC3->6->12-> DC3 | 3.1174 | |

(1) Results under uncertain demand

To verify the effectiveness of the robustness measure to adjust the conservative level of the model, the adjusted value is used to show the trend of the total cost of the objective function under different demand fluctuations and the difference between the location scheme and the path planning scheme. The parameters $\Gamma = 0, 0.4, 0.8$, and 1 are selected in turn, which represent the degree of demand fluctuation as demand determination, small demand fluctuation, large demand fluctuation, and maximum uncertainty of demand fluctuation. Corresponding to the market product classification, it can be divided into four scenarios: market stable products, market demand generally stable products, market demand fluctuation products, and market demand extremely unstable products. The optimization results under different values of $\Gamma$ are shown in Table VI.

TABLE VI
COMPARISON OF THE RESULTS UNDER DIFFERENT $\Gamma$ VALUES.

| $\Gamma$ | Cost of each part (yuan) | Total cost (yuan) | Distribution center selection | Total transport volume (t) |
|---|---|---|---|---|
| 0 | 5950.0000 3795.4259 2962.1365 519.6400 | 13227.2024 | DC1、DC3 | 19.3918 |
| 0.4 | 5950.0000 3993.2922 3145.2619 519.6400 | 13608.1941 | DC1、DC3 | 20.3843 |
| 0.8 | 5950.0000 4191.1585 3328.3874 519.6400 | 13989.1859 | DC1、DC3 | 21.3768 |
| 1 | 5950.0000 4279.8785 3456.7237 526.2875 | 14212.8897 | DC1、DC3 | 21.8763 |

As the $\Gamma$ value increases, the fluctuation of demand also increases, and the degree of uncertainty risk considered by the model increases. When the boundary value of $\Gamma$ is 1, the uncertainty is considered the most in the constraint, and the obtained decision scheme is the most conservative. It can be seen from Table VI that the total transportation volume of the product increases with the increase of the value $\Gamma$. Based on the original intention of robust optimization, the resource allocation scheme obtained by the model is feasible in any state within the uncertainty set. Therefore, in order to overcome the increasing interference of uncertain factors of demand and ensure the quality of enterprise services, the supply also needs to be increasing to ensure the satisfaction of demand, and the cost of cold chain logistics increases gradually. The trend of cost change is as shown in Fig. 4.

With the increase of uncertainty, to meet the needs of consumers to the greatest extent and overcome the risks brought by uncertainty, the minimum cost of system integration is increasing. When $\Gamma = 0.4$, the cost increases to 13608.1941 yuan. As the uncertainty disturbance increases, the cost increases. When $\Gamma = 1$, the demand fluctuation is considered to the greatest extent, and the uncertainty disturbance is maximized. From Fig. 4, it can be seen that the

transportation cost f2 and the cargo damage cost f3 change most obviously. The transportation cost increases with a growth rate of 12.76%.

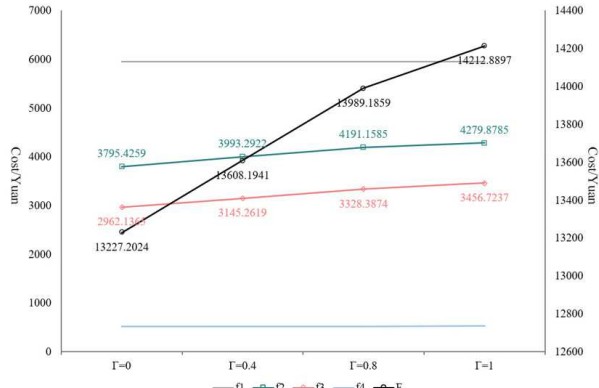

Fig. 4. Costs under different $\Gamma$ values

The cost of cargo damage increased from 2962.1365 yuan ($\Gamma = 0$) to 3456.7237 yuan ($\Gamma = 1$), with a growth rate of 16.69%. The fluctuation of location cost and time window penalty cost is small. When considering the different conservatism of the model, the fluctuation of demand will be different. In order to meet the demand constraints, the model will increase the distribution volume of fresh products, so as to ensure that it can meet the changing needs of customers.

However, there is a significant positive correlation between transportation cost and cargo damage cost and product transportation volume. Therefore, when the demand fluctuation increases, the change of transportation cost and cargo damage cost is the most obvious. It can be seen from Table VI that the choice of distribution center does not change under different $\Gamma$ values, and the first and third distribution centers are selected. The path planning under different $\Gamma$ values is shown in Fig. 5.

As shown in Fig. 4 and Fig. 5, when the $\Gamma$ value changes, the cost changes significantly. However, the planning scheme of the transportation path does not change. Lines with different colors represent the path passed by a car. Since the text assumes that only one type of vehicle participates in the distribution, it can be concluded from the diagram that in each case, six vehicles are distributed, and the distribution path does not change. Path planning is S- > DC1- > 5-4- > DC1, S- > DC1- > 3- > 7- > DC1, S- > DC1- > 11- > 2- > DC1, S- > DC1- > 1- > 8- > DC1, S- > DC3- > 6- > 12- > DC3, S- > DC3- > 10- > 9- > DC3. In order to overcome the increase of interference caused by demand fluctuation and meet the demand of consumption points to the greatest extent, the distribution volume will be adjusted accordingly, which will gradually increase the logistics cost.

In addition, the distribution region of the optimal solution can be controlled by adjusting the $\Gamma$ value, which verifies that the robust measure can effectively adjust the robust conservatism level of the model.

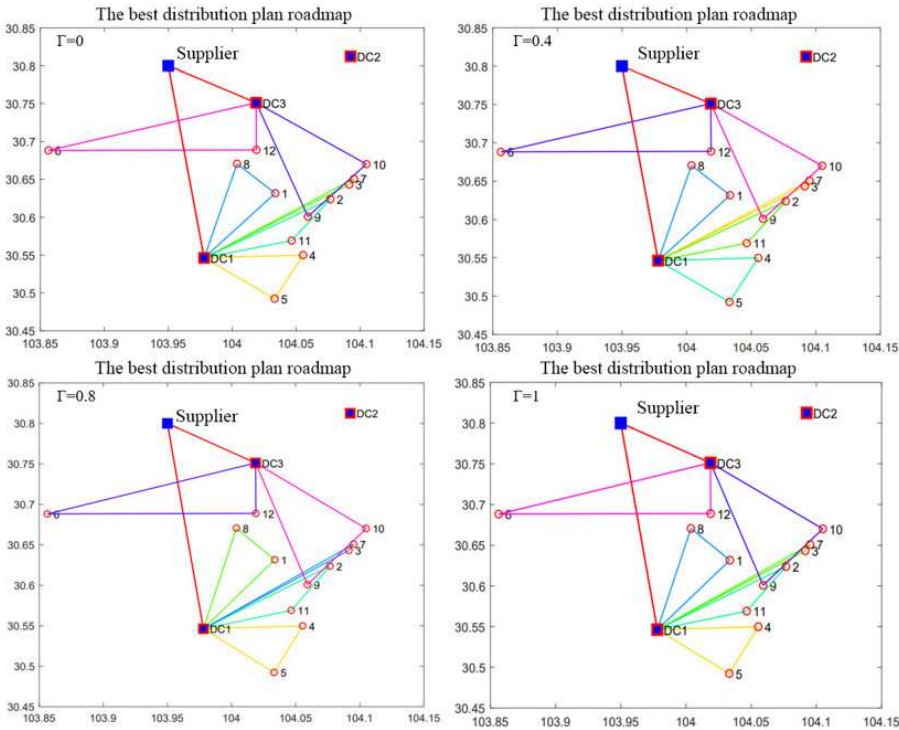

Fig. 5. Logistics path planning under different Γ values.

Therefore, the fluctuation of market demand will have a great impact on the decisions made by managers. Generally conservative managers tend to take market risk into account to a greater extent, that is, the Γ value is greater. In order to meet the needs of consumers to the greatest extent, they tend to increase the transportation volume of fresh products. At this point, Γ = 0.8 can be applied to make a decision. Risk managers tend to minimize the market risk as much as possible, and think that the demand disturbance is small and the market is relatively stable. At this time, they can make decisions with reference to the data support of Γ = 0.4. At the same time, from another point of view, different model conservatism can be applied to different products. For some products with stable market demand and consumers' regular ordering, the decision can be made with reference to the scheme formulated by the optimization model when the demand is constant (Γ = 0).

However, for some consumers with irregular orders and emerging products in the market, the decision-making scheme obtained by the more conservative model can be considered for decision-making. In summary, considering the volatility of market demand plays an important role in the decision-making of cold chain logistics network planning.

*C. Sensitive Analysis*

Due to the influence of weather, urban traffic conditions, market oil price fluctuations and uncertain factors in the transportation process, the unit freight will show certain fluctuations. The change of unit freight will have a great impact on the cost of the whole system and the planning scheme of the cold chain logistics network. In order to explore the influence of the change of unit freight rate on the optimization scheme, under the condition of certain demand, this chapter takes the average freight rate $C_t=3$ as the standard, and analyses the sensitivity of unit freight rate with a

fluctuation range of 25% and a fluctuation range of 50% in the upper and lower limits to represent the five scenarios under different unit freight rates. The resulting cost changes are shown in Fig. 6.

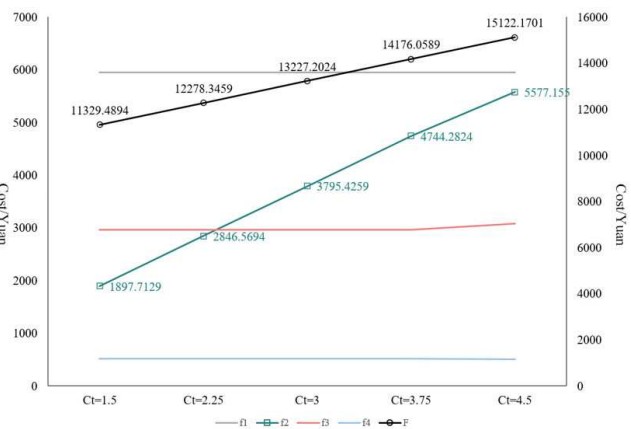

Fig. 6. Cost changes under freight changes.

As shown in Fig. 6, the total cost of the system changes significantly under different unit freight rates. As the unit freight rate increases from 1.5 yuan/ton/km to 4.5 yuan/ton/km, the comprehensive cost of the system are on the rise. By decomposing the costs, it can be seen that the transportation cost has changed the most, with a growth rate of 193.89%. However, other costs have not changed much, which shows that the change of unit freight has little effect on transportation volume. However, when the unit transportation cost changes from 3.75 yuan to 4.5 yuan, the cost of cargo damage increases, which indicates that if the unit transportation cost is too large, it will have a certain impact on the transportation volume, thus indirectly causing the cost of cargo damage to change. The transportation cost is a large part

of the total cost of the system, and it is also the key research object in the cost optimization of the enterprise. From the sensitivity analysis, it can be seen that the effective control of the unit transportation cost not only reduces the transportation cost, but also reduces the cargo damage cost. It can achieve a certain degree of improvement and have a significant impact on the improvement improving the efficiency of the whole enterprise.

Therefore, in the cost optimization, enterprises can achieve transportation cost control by improving transportation efficiency and combining high-tech such as information sharing, so as to optimize the comprehensive benefits of the whole enterprise. Under different unit freight conditions, its path planning has also undergone some changes.

In Fig. 7, the left side of the path planning diagram is the transportation path planning scheme formed by the unit freight cost in the range of 1.5 yuan/ton/km to 3.75 yuan/ton/km. In this range, the unit transportation cost changes, but the transportation path does not change at this time, and the change of unit freight only has a large impact on the transportation cost. However, when the unit transportation cost increases to 4.5 yuan/ton/km, the planned path plan has undergone subtle changes. Compared with the previous scheme, the demand points of DC3 service in the distribution center are 9 and 10. When the freight rate increases to CNY 4.5, the demand is satisfied by DC1. The demand point 8, which was previously satisfied by DC1, is now satisfied by DC3. It can be seen that when the unit freight rate changes in a small range and the cost is low, the price fluctuation only affects the transportation cost, and the corresponding impact on the total cost of the system. However, when the cost increases to a higher level, in order to ensure the satisfaction of consumer demand, the path planning scheme will also make appropriate adjustments. Therefore, when the freight price is high, not only the cost of the enterprise will increase accordingly, but also it is necessary to change the transportation path and update the previous transportation planning strategy according to the actual situation. Therefore,

reasonable control of unit transportation costs and prevention of excessive unit costs can not only directly reduce transportation costs and cargo damage costs in the cold chain system, but also avoid re-planning of transportation paths and save transportation strategic planning costs.

### D. Method Comparison

Besides, we have considered using an available method (the greedy algorithm) to observe and compare the corresponding results, to verify the effectiveness of the method proposed in this paper. The comparison results are shown in the table VII.

As indicated in Table VII, when the model is deterministic, The performance of the genetic algorithm is better than that of greedy algorithm. Specifically, the genetic algorithm yields results with lower costs and a reduced total transport volume, suggesting its effectiveness in minimizing fresh product loss during transportation and lowering operational expenses for the enterprise. However, it is noteworthy that the costs incurred in the robust model surpass those in deterministic model. This disparity arises from the prioritization of model stability over immediate benefits. Nevertheless, this comparison underscores the validity and practical applicability of our model.

TABLE VII
METHOD COMPARISON.

| Model | Algorithm | Total cost (yuan) | Distribution center selection | Total transport volume (t) |
|---|---|---|---|---|
| Deterministic model | Greedy | 14420.271 | DC1、DC3 | 23.7726 |
| | Genetic | 13227.2024 | DC1、DC3 | 19.3918 |
| Rubust model | Genetic | 13608.1941 | DC1、DC3 | 20.3843 |

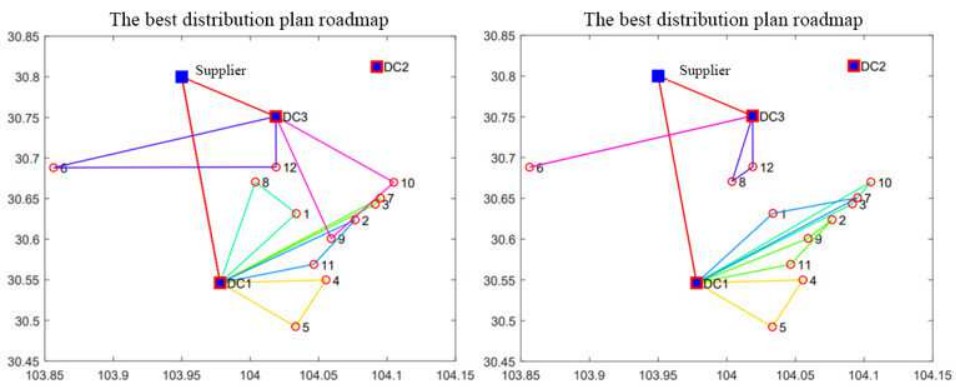

Fig. 7. Path planning under different freight rates.

### V. CONCLUSIONS

In this paper, a robust optimization method is introduced to solve the influence of uncertain factors. Taking the minimum total cost of the logistics system as the objective function, the cost consists of four parts : the location cost of the distribution center, the transportation cost, the cargo

damage cost of fresh products and the penalty cost of violating the time window. Considering several constraints such as capacity constraints, vehicle volume constraints, and demand satisfaction, a robust optimization model is constructed, and the solution steps of the model are designed based on genetic algorithm. And the model is applied to the case A enterprise for simulation. By considering the results of different

conservative degrees, the scheme of distribution center location and optimal path planning is obtained. According to the results, in transportation, enterprises must first plan the transportation paths scientifically and rationally, monitor the transportation process by applying high technology, and scientifically judge the transportation conditions to make optimal planning of transportation paths, so as to achieve the purpose of controlling transportation costs. Second, choose a reasonable mode of transportation, according to the characteristics of the product, the volume of transportation and the real-time situation of urban traffic, choose the most reasonable mode of transportation for product distribution, from the means of transportation to reduce the unit transportation cost. In addition, the utilization rate of vehicle return is fully considered to reduce the waste of capacity caused by idle on the road. Finally, increase the number of vehicle turnover in the cycle. In a certain cycle, the vehicle will always have a fixed expenditure. In this cycle, increase the number of vehicle turnover, and allocate the fixed cost to each transportation. The transportation cost will be reduced to reduce the transportation cost.

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
