# OpenReview forum: "$Robust Optimization of Cold Chain Logistics Networks with Time Window under Uncertain Demand: A Case Study in China$"
_IEEE.org/ICIST/2024/Conference — IEEE ICIST 2024 Conference Submission_

### Official Review · Reviewer_y24v · 2024-08-25
**Robust Optimization of Cold Chain Logistics  Networks with Time Window under Uncertain  Demand: A Case Study in China**

**Rating:** 8
**Confidence:** 3

**Review:**

The research content of this paper is innovative, considering a practical need by optimizing the cold chain network system. The proposed optimization scheme was validated through simulation, achieving the goal of minimizing the overall system costs. However, some minor issues still need to be improved.

---

### Official Review · Reviewer_JxS1 · 2024-09-01
**Comments to paper 55**

**Rating:** 9
**Confidence:** 5

**Review:**

Considering the high uncertainty of cold chain market demand, this paper adopts the robust optimization method to construct an optimization model of cold chain logistics network with a time window by taking the minimization of the comprehensive cost of the system as the goal, and uses genetic algorithm to solve the model. Some comments should be considered.
1. Comparative results are necessary to validate the claimed advantages of the proposed approach.
2. Conference paper exceeds the page limit.
3. The font of the image is not clear.
4. Some typos and grammar errors should be modified in this paper. Please carefully doublecheck the manuscript.

---

### Official Review · Reviewer_VPcG · 2024-09-01
**This paper constructs an optimization model of cold chain logistics network with a time window by taking the minimization of the comprehensive cost of the system as the goal, and uses genetic algorithm to solve the model.**

**Rating:** 6
**Confidence:** 5

**Review:**

1.It is recommended that authors summarize the contribution of the method proposed in this paper to highlight the research advantages of this paper.
2.How to address the issues of local optima and nonlinear constraints in the optimization model of cold chain logistics network with a time window.
3.Formulas (16) and (20) require adjustment.
4.Figures 1-figure 7 are unclear.
5.It is suggested to include a comparison with other methods in the simulation section.

---

### Decision · Program_Chairs · 2024-09-06

Accept (Oral)